# Isolation of the *Buchnera aphidicola* flagellum basal body complexes from the *Buchnera* membrane

**Matthew J. Schepers[1], James N. Yelland[1], Nancy A. Moran [2]\*, David W. Taylor [1,3,4,5]\***

**1** Institute for Cell and Molecular Biology, University of Texas at Austin, Austin, Texas, United States of America, **2** Department of Integrative Biology, University of Texas at Austin, Austin, Texas, United States of America, **3** Departmnet of Molecular Biosciences, University of Texas at Austin, Austin, Texas, United States of America, **4** Center for Systems and Synthetic Biology, University of Texas at Austin, Austin, United States of America, **5** LIVESTRONG Cancer Institute, Dell Medical School, Austin, Texas, United States of America

\* dtaylor@utexas.edu (DWT); nancy.moran@austin.utexas.edu (NAM)

## Abstract

*Buchnera aphidicola* is an intracellular bacterial symbiont of aphids and maintains a small genome of only 600 kbps. *Buchnera* is thought to maintain only genes relevant to the symbiosis with its aphid host. Curiously, the *Buchnera* genome contains gene clusters coding for flagellum basal body structural proteins and for flagellum type III export machinery. These structures have been shown to be highly expressed and present in large numbers on *Buchnera* cells. No recognizable pathogenicity factors or secreted proteins have been identified in the *Buchnera* genome, and the relevance of this protein complex to the symbiosis is unknown. Here, we show isolation of *Buchnera* flagellum basal body proteins from the cellular membrane of *Buchnera*, confirming the enrichment of flagellum basal body proteins relative to other proteins in the *Buchnera* proteome. This will facilitate studies of the structure and function of the *Buchnera* flagellum structure, and its role in this model symbiosis.

## Introduction

*Buchnera aphidicola* is an obligate endosymbiont of aphid species worldwide [1] and is a model for bacterial genome reduction, maintaining one of the smallest genomes yet discovered, only 600 kbps [2, 3]. Though *Buchnera* has lost genes not essential for its symbiotic lifestyle [2, 4, 5] it retains genes associated with amino acid biosynthesis, reflecting its participation in a nutritional symbiosis [2, 6, 7]. Though the exchange of amino acids and vitamins between the aphid host and *Buchnera* has been well-documented [6, 8, 9], the molecular mechanism for how these metabolites cross *Buchnera* membranes is unknown: *Buchnera* maintains a small number of genes coding for membrane transport proteins, most of which are located at the inner membrane (MdlAB, PitA, YggB, ZnuABC) [2, 10]. The permeability of the *Buchnera* outer membrane remains a mystery, considering the paucity of annotated transporter genes in sequenced *Buchnera* genomes. Genes coding for proteins localizing to the outer membrane of *Buchnera* include small β-barrel aquaporins (OmpA, OmpF), which allow

Consortium via the PRIDE35 partner repository with the dataset identifier PXD024664 and 10.6019/PXD024664.

**Funding:** This work was supported by the National Science Foundation 1551092 (to N.A.M), a Welch Foundation grant F-1938 (to D.W.T.), National Institute of General Medical Sciences (NIGMS) of the National Institutes of Health (NIH) R35GM138348 (to D.W.T.), Army Research Office Grant W911NF-15-1-0120 (to D.W.T.), and a Robert J. Kleberg, Jr. and Helen C. Kleberg Foundation Medical Research Award (to D.W.T.). D.W.T is a CPRIT Scholar supported by the Cancer Prevention and Research Institute of Texas (RR160088) and an Army Young Investigator supported by the Army Research Office (W911NF-19-1-0021). The funders had no role in study design, data collection and analysis, decision to publish, or preparation of the manuscript.

**Competing interests:** The authors have declared that no competing interests exist.

passive diffusion of small molecules, and flagellum basal body components (FlgBC/FGH/K) [2, 9, 10]. Investigation into protein expression by these symbiotic partners has shown that flagellum basal body components are highly expressed by *Buchnera* [11]. Indeed, transmission electron microscopy images of *Buchnera* reveal flagellum basal bodies studded all over the bacterial outer membrane [12]. Despite its abundance on the *Buchnera* cell surface, the role of this protein complex for maintaining the aphid-*Buchnera* symbiosis is unknown [13].

*Buchnera* of the pea aphid (*Acyrthosiphon pisum*) maintains 26 genes coding for flagellum-related proteins (Fig 1).

These 26 genes, located on the chromosome in three discrete clusters, code for the structural proteins required for formation of a flagellum basal body (*fliE, fliF, flgB, flgC, flgF, flgG,* and *flgH*), a partial flagellar hook (*flgD, flgE, flgK*), as well as the Type III cytoplasmic export proteins (*flhA, flhB, fliP, fliQ,* and *fliR*). *Buchnera* lineages vary in the set of flagellum genes retained (S1 Table), but all have lost genes encoding the flagellin and motor proteins (*fliC, motA, motB*) [14], indicating a functional shift away from cell motility. The bacterial flagellum structure is an evolutionary homologue to the injectisome (Type III secretion system, or T3SS), a macromolecular protein complex used to deliver secrete effector proteins, often to a eukaryotic host [15–17]. Flagellum assembly occurs in a stepwise, sequential manner beginning from the bacterial cytoplasm, identical to the T3SS [18–20]. *Buchnera* maintains genes coding for the proteins required for a functional T3SS [2, 12], as shown in studies of *Yersinia* [21], and *Salmonella* [22, 23]. Gram-negative bacteria have also been shown to export proteins through a flagellum basal body [21, 24, 25]. The bacterial flagellum could be repurposed to serve a novel function for the aphid-*Buchnera* symbiosis. The basal body could serve as a type III protein exporter to secrete proteins to signal to the aphid host or as a surface signal molecule for host recognition during infection of new aphid embryos. Here, we present a procedure for isolation of flagellum basal body complexes adapted for an endosymbiont [26], allowing for removal of these structures directly from *Buchnera* and enrichment of flagellum basal body complexes after isolation. This procedure will enable further characterization of the basal bodies and their modifications for a role in symbiosis.

## Results

### Isolation of hook basal body complexes from *Buchnera*

Purification of the complex was initially assessed at multiple timepoints along the procedure. Samples were taken of initial *Buchnera* cell lysate, lysate after raising the pH to 10, protein suspension after the first 5000g spin, the third 5000g spin, and finally after the 30,000g spin and overnight incubation in TET buffer. SDS-PAGE showed sixteen bands were present after the staining procedure and their sizes corresponded to those of constituent proteins of the *Buchnera* flagellum basal body (S1 Fig). Protein samples were extracted from the gel and subjected to mass spectrometry analysis.

### Mass spectrometry analysis of isolated basal body complexes

Protein ID LC-MS/MS spectral counts were provided by the University of Texas at Austin Proteomics Core Facility. We compared our samples to proteomic datasets from homogenized whole aphids, and from bacteriocytes purified from pea aphids [11]. *Buchnera* flagellum-related proteins were highly enriched by our isolation procedure, especially FliF, FlgI, FlgE, FlhA, and FlgF (Fig 2).

These results indicate that all but two flagellum proteins from *Buchnera* cells present in the mass spectrometry samples were enriched during the isolation procedure: structural proteins FilE, FliF, FlgI, FlgE, FlgF, and FlgH were enriched threefold or more from the start to the

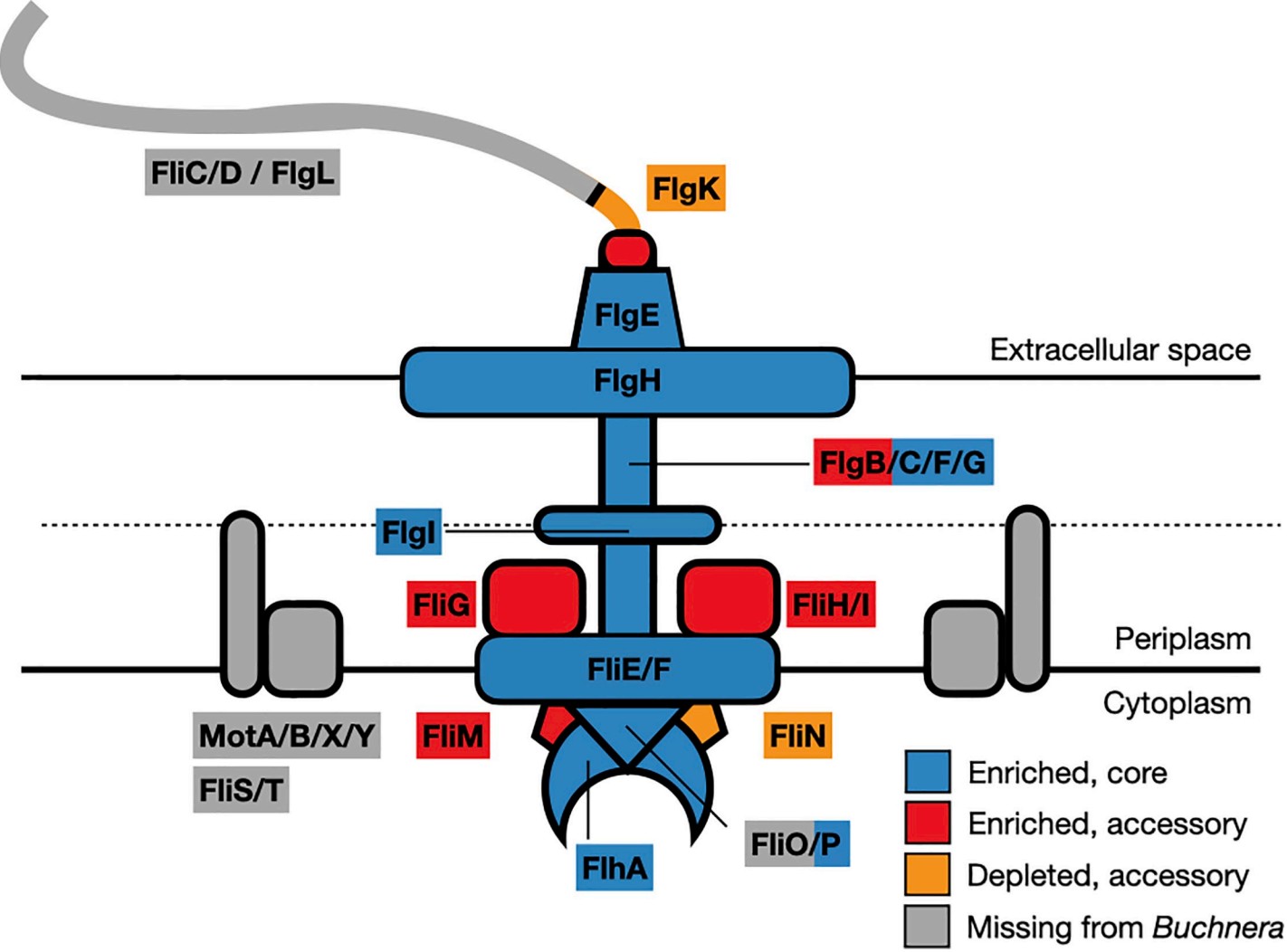

**Fig 1. Cartoon diagram of the bacterial flagellum.** Colors indicate which proteins are lost from *Buchnera aphidicola* of pea aphids and also the enrichment status of each retained protein at the final step of the procedure (Fig 2).

finish of the procedure. FlgB, FlgC, FlgG, FliG, FliH, and FliI were enriched, though not to the extent of the other structural proteins. Type III secretion proteins FlhA and FliP were shown to be enriched by this procedure (Fig 1, S2 Fig). GroL and Tuf, the two most abundant proteins expressed by *Buchnera* [11], were both depleted at the end of the procedure. The widespread enrichment of *Buchnera* flagellum-related proteins indicate that our adapted procedure for isolating macromolecular protein complexes from the membranes of endosymbiotic bacteria was successful. Only two flagellum-related proteins, FlgK and FliN, were reduced by the isolation procedure, perhaps because of their localization to the periphery of the flagellum basal body complex.

## Basal bodies remain partially intact when observed under electron microscopy

We analyzed the isolated basal bodies by negative stain electron microscopy. While raw micrographs showed heterogenous particles, likely due to disassembly of the complex, detergent

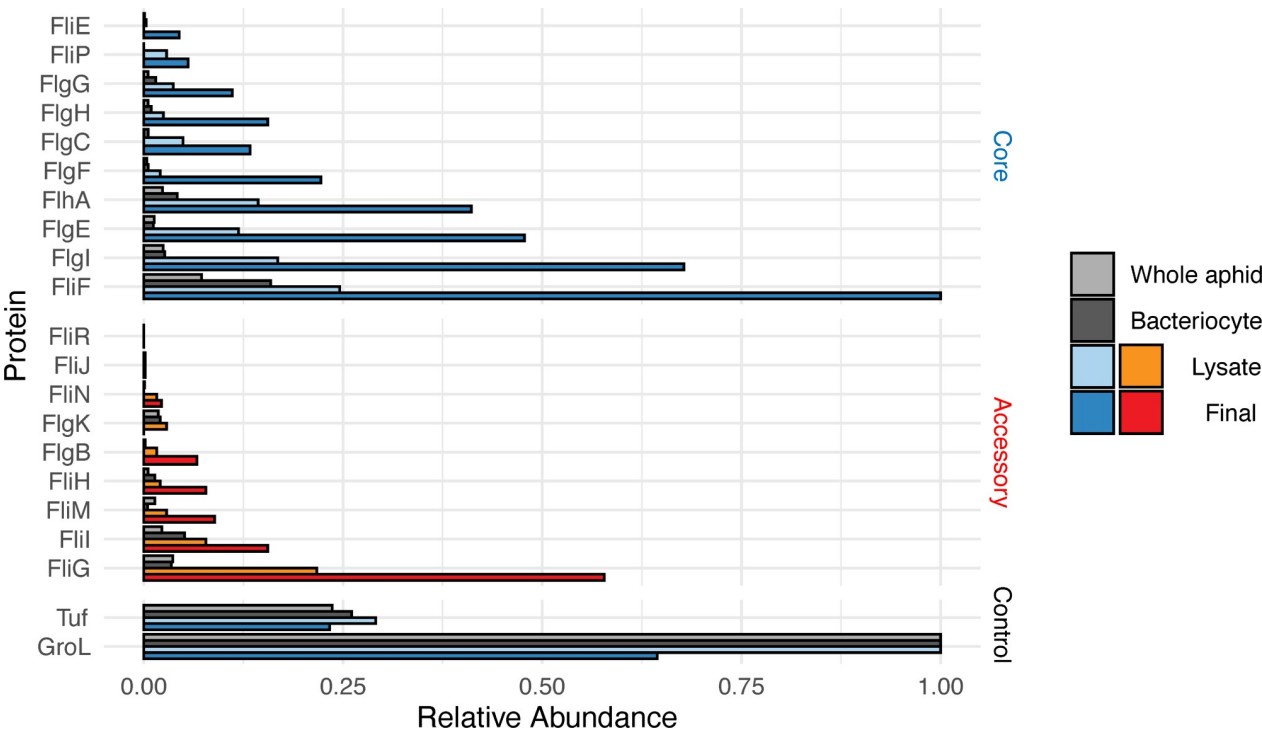

**Fig 2. Barplot showing enrichment of flagellum-related proteins before (Lysate) and after (Final) the isolation procedure.** Results from the protein enrichment procedure are compared to proteomic datasets generated with whole aphids and dissected bacteriocytes. Blue indicates "core" proteins required for secretion activity and red indicates accessory proteins maintained by *Buchnera aphidicola* in pea aphids.

micelles, and contaminating proteins, there were several particles that appeared regularly. These single particles resembled characterized flagellum basal body structure, with both rod and ring-shaped features (Fig 3), similar in size and shape to TEM images of basal body complexes on membranes of whole *Buchnera* cells from Maezawa *et al* [12], indicating success in isolating intact, or mostly intact, basal body complexes. Of 204 micrographs, 24 contained at least one easily discernible, partially intact basal body complex.

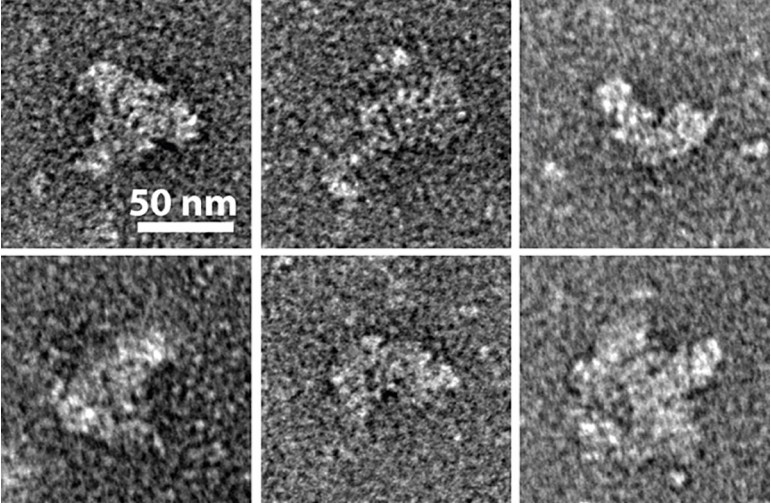

**Fig 3. Single particles of *Buchnera* flagellum complexes after the isolation procedure.** Scale bars represent 50nm.

## Discussion

Here, we demonstrate a procedure for isolating macromolecular protein complexes from *Buchnera aphidicola*, an obligate endosymbiotic bacterium that cannot be cultured or genetically manipulated. Identifying the protein losses in these complexes compared to free-living, motile *Enterobacteria* could elucidate how *Buchnera's* proteome has been affected by adaptation over millions of years to a mutualistic lifestyle.

As *Buchnera* is not motile and is confined to host-derived "symbiosomal" vesicles inside bacteriocytes [27, 28], the retention and expression of these partial flagellar structures indicates that they have become repurposed. These complexes have previously been hypothesized to be acting as type III secretion systems for provisioning peptides or signal factors to the aphid host [13]. Indeed, the proteins retained in the *Buchnera* flagellum-related structure constitute the structural proteins and machinery required for a functional type III secretion system [21]. Transcriptome analyses of pea aphid lines with different *Buchnera* titers reveal differences in expression of flagellar genes [29]. In aphid lines that harbor low titer populations of *Buchnera*, the endosymbionts have elevated relative expression of mRNA associated with flagellar secretion genes (*fliP*, *fliQ*,and *fliR*), while *Buchnera* in aphid lines with high *Buchnera* titers had elevated expression of genes for flagellum structural proteins [29].

Though heavily expressed in *Buchnera* of pea aphids, components of the flagellum basal body are not maintained equally among lineages of *Buchnera* of different aphid species based on available genomic sequences [14] (S1 Table). Genes coding for proteins associated with type III secretion activity (*flhA*, *flhB*, *fliP*, *fliQ, and fliR)* and basal body structural proteins (*fliE, fliF, flgB, flgC, flgF, flgG,* and *flgH)* are well maintained across *Buchnera* lineages, but genes coding for hook proteins (*flgD*, *flgE*, and *flgK)* and the flagellum-specific ATPase (*fliI)* are frequently shed. In all but the most extreme examples, the *Buchnera* flagellum-related genes are retained and intact, pointing to a continuing role for this complex for this ancient symbiosis. However, the *Buchnera* strain harbored by aphids of genus *Stegophylla*, which has the smallest sequenced *Buchnera* genome discovered thus far (412 kbps), has completely lost genes associated with the flagellum structure and Type III secretion activity.

Beyond *Buchnera*, retention of the flagellar apparatus varies among obligate endosymbionts. Considering other symbionts in the Gammaproteobacteria, the complex is sometimes completely intact, as in the tsetse fly symbiont *Wigglesworthia glossinidius*, which expresses this system and is motile only when infecting progeny within the mother's body [30]. The symbiont of armored snails resembles *Buchnera* of pea aphid in retaining intact genes for the basal body complex while lacking intact *fliC*, encoding flagellin [31]. But all flagellum-related genes are lost from genomes of some insect symbionts, including the genera *Baumannia* and *Blochmannia*, symbiotic in cicadas and carpenter ants respectively.

*Buchnera's* tiny genome contains no known pathogenicity proteins or proteins previously associated with type III export [2, 32]. Potentially, *Buchnera* flagellum basal bodies may instead serve as surface signals for recognition by the host. Vertical transfer of *Buchnera* from mother to daughter aphids shows naked *Buchnera* cells being exocytosed from maternal bacteriocytes and passing through aphid haemolymph to infect a nearby specialized syncytial cell of stage 7 embryos [33]. The purpose of the flagellum basal body complex in the context of *Buchnera's* symbiotic lifestyle remains unknown. Further inquiry into this protein complex could reveal the repurposing of a motility organelle for driving this ancient and obligate symbiosis.

## Methods

A step-by-step protocol is provided as S1 Protocol. This protocol is also available at dx.doi.org/10.17504/protocols.io.bs5wng7e.

## *Buchnera* extraction from aphids

Pea aphids (*Acrythosiphon pisum* strain LSR1) were grown as an all-female clone on Fava bean (*Vicia faba)* seedlings on 16h/8h light/dark cycles at 20˚C. Once reaching adulthood, apterous adults were raised on Fava bean plants on 16h/8h light cycles and allowed to reproduce over-night. After seven days, all aphids, in the 4th larval instar and typically amounting to 5g, were removed from the Fava bean plants. Aphids were weighed and surface-sterilized in 0.5% NaClO solution, then rinsed twice in Ultrapure water (MilliporeSigma), each 30 s. Aphids were gently ground in a mortar and pestle in 40mL sterile Buffer A (25mM KCl (Sigma-Aldrich), 35mM Tris base (Sigma-Aldrich), 10mM $MgCl_2$ (Sigma-Aldrich), 250mM anhy-drous EDTA (Sigma-Aldrich), and 500mM Sucrose (Sigma-Aldrich) at pH 7.5). Aphid homogenate was vacuum filtered to 100μm, then centrifuged at 1500g for 10 min at 4˚C. Supernatant was discarded, and the resulting pellet was resuspended in 20mL Buffer A and vacuum-filtered three times from 20μm, to 10μm, and finally to 5μm. The resulting filtrate was spun at 1500g for 30 min at 4˚C and supernatant discarded. The resulting pellet was resus-pended in 10mL Sucrose solution (300mM sucrose (Sigma-Aldrich) and 100mM Tris base (Sigma-Aldrich)) then checked on a brightfield microscope for intact *Buchnera* cells. *Buchnera* cells remain intact while at 4˚C for a maximum of 24h.

## Isolation of flagellum basal bodies from *Buchnera* cells

*Buchnera* was incubated with gentle spinning on ice with egg white lysozyme (0.1mg/mL, Sigma-Aldrich) for 30m. 100mM Anhydrous EDTA solution, pH 7.5 (Sigma-Aldrich) was added to final concentration 10mM. The pellet was taken off ice, and gradually raised to room temperature with gentle spinning for 30 min. Triton X-100 (Acros Organics) was added to 1% w/v, along with 1mg/mL RNase-free DNase I (Bovine Pancreas, Sigma-Alrich) and allowed to stir for 30 min. After incubation, cell lysate was kept at 4˚C or on ice until use. The lysate was raised to pH 10 using 1N NaOH (Macron Fine Chemicals) to attempt to denature host and bacterial cytoplasmic proteins. The solution was spun at 5000g for 10 min at 4˚C three times, each time decanting the supernatant to a new tube. After three spins, the supernatant was transferred to a Nalgene Oak Ridge polyallomer centrifuge tube (Thermo-Fisher) and spun at 30,000g for 1 h at 4˚C. Supernatant was gently decanted and pellet covered with TET buffer (10mM Tris-HCl, 5mM EDTA, 0.1% X-100, pH 8.0) and left overnight at 4˚C to soften and dissolve.

## Submission of protein for mass spectrometry

Solubilized protein concentration was determined using an Eppendorf Biophotometer. 1.5mg protein was run on premade 4–12% Tris-Glycine SDS-PAGE gels (Thermo-Fisher) at 120V for 10m. Gels were stained in Coomassie Brilliant Blue (Bio-Rad) for 30 min, then destained in 20% acetic acid (Thermo-Fisher) for 30m. Gel bands corresponding to the step in the proce-dure sampled ("Lysate," "pH 10," "Spin 1," "Spin 3," "Final") were cut out and submitted to the University of Texas at Austin CBRS Biological Mass Spectrometry Facility for LC-MS/MS using a Dionex Ultimate 3000 RSLCnano LC coupled to a Thermo Orbitrap Fusion (Thermo-Fisher). Samples were submitted in 50mL destain with *Buchnera aphidicola* str. APS (ASM960v1) provided as the reference organism. Prior to HPLC separation, peptides were desalted using Millipore U-C18 ZipTip Pipette Tips (Millipore-Sigma). A 2cm long x 75μm ID C18 trap column was followed by a 25cm long x 75μm analytical columns packed with C18 3μm material (Thermo Acclaim PepMap 100, Thermo-Fisher) running a gradient from 5–35%. The FT-MS resolution was set to 120,000, with an MS/MS cycle time of 3 s and acquisi-tion in HCD ion trap mode. Raw data were processed using SEQUEST HT embedded in

Proteome Discoverer (Thermo-Fisher). Scaffold 4 (Proteome software) was used for validation of peptide and protein IDs. The mass spectrometry proteomics data have been deposited to the ProteomeXchange Consortium via the PRIDE [34] partner repository with the dataset identifier PXD024664 and 10.6019/PXD024664.

### EM and data collection

Protein from the final step of this procedure was stained using 3% Uranyl Acetate on a 400-mech continuous carbon grid. Images were acquired using an FEI Talos transmission electron microscope operating at 200 kV, with 1.25 s exposures, a dose rate of 19 e- Å-2, and a nominal magnification of x57000.

### Whole aphid proteomic samples

For controls, proteomes were profiled for whole aphids, including both *Buchnera* and aphid cells. Aphids were mixed-aged populations grown at 20˚C in 30 cup cages and pooled into three replicate samples. Aphids were washed and homogenized in buffer as described above. The homogenate was centrifuged at 4000g for 15min at 4˚C, Supernatant was removed, and pellet was suspended with 2% SDS, 0.1M Tris-HCl, 0.1M DTT at 100˚C for 10 min, then centrifuged at 14,000g for 20min at 4˚C to remove non-soluble material after adding same volume of 8M Urea. Protein concentration was determined on an Eppendorf BioPhotometer. 5mg total protein was run on a Bis-Tris gel for less than 1 cm, and the band was excised and sent to the UT Proteomics Core for LC-MS/MS protein ID. Protein ID methods were identical as detailed above.

### Supporting information

**S1 Fig. Silver stained SDS gel created after the enrichment procedure was performed.** The first lane is taken directly from the enrichment preparation after overnight incubation with TET buffer. The second lane is after concentrating the enriched proteins to 1 mg/mL. The third lane is concentrated protein diluted to 0.5 mg/mL. Ladder values represent molecular weight in kDa. Symbols correspond to flagellar protein molecular weight: * corresponds to FlhA (78 kDa). † corresponds to FliF (63 kDa) and FlgK (63 kDa). ˚ corresponds to FlgE (45 kDa), FliP (43kDa), and FlgI (41 kDa). ‡ corresponds to FliG (38kDa) and FliM (37 kDa). Δ corresponds to FlgG (28 kDa), FlgF (28 kDa), FlgH (26 kDa), and FliH (26kDa). Ø corresponds to FlgB (16 kDa), FliN (15 kDa), FlgC (15 kDa), and FliE (11 kDa).
(TIF)

**S2 Fig. Dotplot of *Buchnera aphidicola* flagellum proteins found after LC/MS-MS analysis.** The enrichment score for each protein is indicated on the x axis. Enrichment scores are calculated by dividing unique spectral counts for each protein in the final step by each protein present in the cell lysate. Core flagellum proteins (defined by proteins required for type III secretion activity and flagellum structure) are filled in green, accessory proteins are filled in white.
(TIFF)

**S1 Table. Flagella genes retained across *Buchnera*, sorted by host aphid species and tribe.** If genes are retained, the gene length in nucleotides is noted.
(XLSX)

**S1 Protocol. Step-by-step procedure for isolation flagellum basal bodies.**
(DOCX)

**S1 Original image.**
(TIF)

## Acknowledgments

We thank Eric Verbeke, Jack Bravo, and Evan Schwartz for their advice and ideas for isolating and imaging proteins from native cells; Julie Perreau, Margaret Steele, and Serena Zhao for creating a space in which ideas and techniques could be shared freely; Kim Hammond for help with aphid raising and organization. Thomas Smith was the first to attempt the FBB extraction experiments helped develop the protocols and ideas.

## Author Contributions

**Conceptualization:** Nancy A. Moran, David W. Taylor.

**Data curation:** Matthew J. Schepers, James N. Yelland.

**Formal analysis:** Matthew J. Schepers, Nancy A. Moran, David W. Taylor.

**Funding acquisition:** Nancy A. Moran, David W. Taylor.

**Investigation:** Matthew J. Schepers.

**Methodology:** Matthew J. Schepers.

**Supervision:** Nancy A. Moran, David W. Taylor.

**Writing – original draft:** Matthew J. Schepers, Nancy A. Moran, David W. Taylor.

**Writing – review & editing:** James N. Yelland.

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
