## [Decision Letter · Decision Letter 0]

11 Feb 2021

PONE-D-21-00952

Isolation of the Buchnera aphidicola flagellum basal body from the Buchnera membrane

PLOS ONE

Dear Dr. Taylor,

Thank you for submitting your manuscript to PLOS ONE. After careful consideration, we feel that it has merit but does not fully meet PLOS ONE’s publication criteria as it currently stands. Therefore, we invite you to submit a revised version of the manuscript that addresses the points raised during the review process.

We look forward to receiving your revised manuscript.

Kind regards,

Chih-Horng Kuo, Ph.D.

Academic Editor

PLOS ONE

Journal Requirements:

2.PLOS ONE now requires that authors provide the original uncropped and unadjusted images underlying all blot or gel results reported in a submission’s figures or Supporting Information files. This policy and the journal’s other requirements for blot/gel reporting and figure preparation are described in detail at https://journals.plos.org/plosone/s/figures#loc-blot-and-gel-reporting-requirements and https://journals.plos.org/plosone/s/figures#loc-preparing-figures-from-image-files. When you submit your revised manuscript, please ensure that your figures adhere fully to these guidelines and provide the original underlying images for all blot or gel data reported in your submission. See the following link for instructions on providing the original image data: https://journals.plos.org/plosone/s/figures#loc-original-images-for-blots-and-gels.

Reviewers' comments:

Reviewer's Responses to Questions

**Comments to the Author**

1. Is the manuscript technically sound, and do the data support the conclusions?

Reviewer #1: Yes

Reviewer #2: Yes

2. Has the statistical analysis been performed appropriately and rigorously? 

Reviewer #1: N/A

Reviewer #2: N/A

3. Have the authors made all data underlying the findings in their manuscript fully available?

Reviewer #1: Yes

Reviewer #2: No

4. Is the manuscript presented in an intelligible fashion and written in standard English?

Reviewer #1: Yes

Reviewer #2: Yes

5. Review Comments to the Author

Reviewer #1: The paper presents a procedure for isolating proteins for the flagellum basal body of Buchnera, which would facilitate further characterization of this enigmatic protein complex. I believe the paper is important and should be published after some minor edits.

My suggestions are as follows.

Title: Probably "basal body" should be replaced with "basal body proteins" or "basal body complexes". The original wording suggests that basal bodies can be isolated intact, which is not certain at this stage.

p8, l9: "flagella" should be "flagellum basal body proteins". As the authors also describe, Buchnera lacks flagellin, meaning that it lacks flagella.

p8, l10-11: "flagellum proteins" and "flagellum structure" should also be corrected accordingly.

Introduction: In this section, I would like to see protein names and a corresponding figure similar to the current Figure 2.

p9, l19-20: Probably "flagellum proteins" should be "flagellum-related proteins".

p10, l14: Probably "basal bodies" should be replaced with "basal body proteins" or "basal body complexes".

p10, l22: Same as above.

p11, l3: Probably better to add "from Buchnera cells", which would clarify the meaning of "the start" in the following sentence. (Otherwise, it could be misunderstood as whole aphid homogenate.)

p11, l12: The fact that "basal bodies resemble top hats" is not a novel finding. I believe that the authors would like to say instead that they succeeded in obtaining apparently intact (?) basal bodies using the present procedure.

p11, l21: What "changes"?

p13, l9, "bleach solution": Better to show the compound name.

p13, l18: One more parenthesis is required after "Aldrich".

p13, l19, "remain alive": Better to show some evidence or references.

p17: A brief explanation about Tuf and GroL is required.

p18: I believe this kind of figure should be shown in the introduction section.

Reviewer #2: In this manuscript, Schepers and co-authors report a protocol for isolating flagellar bodies from the membrane of Buchnera endosymbionts of aphids. Flagellar genes have been previously shown to be highly expressed in Buchnera and the flagellar basal bodies imaged on the surface of its cells, but the function of these structures for the aphid-Buchnera symbiosis remains unclear. Since the bacterial flagellum is homologous to the T3SS, these structures are often hypothesized to be involved in protein exchange between Buchnera cells and the host-derived (symbiosomal) membrane surrounding Buchnera’s double membrane. The manuscript only reports a protocol for isolating the flagellar basal bodies, so it is short and straightforward. My comments are relatively minor.

(1) Since the manuscript reports a new protocol, I would appreciate if the authors could upload a step-by-step protocol with as many details as possible to a server such as protocols.io (or to FigShare or as a supplementary file).

(2) If possible, please upload the raw mass spec data to a database or share them in any other way. I understand that sharing such data as supplementary material is not very common, but in this case I feel that having the data available would be beneficial to anyone trying to reproduce the protocol, e.g. to properly validate their results.

(3) Figure 1. Were any aphid-derived or non-flagellar Buchnera proteins enriched after the isolation procedure?

(4) Figure 3. Please provide more details about how many grids were visualized and how many top hat-like particles of flagellar complexes were approximately detected by TEM. Were any other macromolecular protein complexes enriched in the sample?

(5) Are there any other endosymbionts described with an incomplete flagellum (or T3SS) similar to the one known from Buchnera?

Please correct 4C to 4ºC throughout the MS.

6. PLOS authors have the option to publish the peer review history of their article (what does this mean?). If published, this will include your full peer review and any attached files.

Reviewer #1: No

Reviewer #2: No

---

## [Decision Letter · Decision Letter 1]

6 Apr 2021

Isolation of the Buchnera aphidicola flagellum basal body complexes from the Buchnera membrane

PONE-D-21-00952R1

Dear Dr. Taylor,

We’re pleased to inform you that your manuscript has been judged scientifically suitable for publication and will be formally accepted for publication once it meets all outstanding technical requirements.

Kind regards,

Chih-Horng Kuo, Ph.D.

Academic Editor

PLOS ONE

Additional Editor Comments (optional):

Congratulations on the nice work! Best, CH

Reviewers' comments:

Reviewer's Responses to Questions

**Comments to the Author**

1. If the authors have adequately addressed your comments raised in a previous round of review and you feel that this manuscript is now acceptable for publication, you may indicate that here to bypass the “Comments to the Author” section, enter your conflict of interest statement in the “Confidential to Editor” section, and submit your "Accept" recommendation.

Reviewer #1: (No Response)

Reviewer #2: All comments have been addressed

2. Is the manuscript technically sound, and do the data support the conclusions?

Reviewer #1: Yes

Reviewer #2: Yes

3. Has the statistical analysis been performed appropriately and rigorously? 

Reviewer #1: N/A

Reviewer #2: Yes

4. Have the authors made all data underlying the findings in their manuscript fully available?

Reviewer #1: Yes

Reviewer #2: Yes

5. Is the manuscript presented in an intelligible fashion and written in standard English?

Reviewer #1: Yes

Reviewer #2: Yes

6. Review Comments to the Author

Reviewer #1: (No Response)

Reviewer #2: The authors have addressed all my comments. I have no further questions or comments. Congratulations.

7. PLOS authors have the option to publish the peer review history of their article (what does this mean?). If published, this will include your full peer review and any attached files.

Reviewer #1: No

Reviewer #2: No

---

## [Editor Report · Acceptance letter]

19 Apr 2021

PONE-D-21-00952R1 

Isolation of the *Buchnera aphidicola* flagellum basal body complexes from the *Buchnera* membrane 

Dear Dr. Taylor:

I'm pleased to inform you that your manuscript has been deemed suitable for publication in PLOS ONE. Congratulations! Your manuscript is now with our production department. 

Kind regards, 

on behalf of

Dr. Chih-Horng Kuo 

Academic Editor

PLOS ONE